# The Social Lives of Free-Ranging Cats

**DOI:** 10.3390/ani12010126

**Published:** 2022-01-05

**Authors:** Kristyn R. Vitale

**Affiliations:** Animal Health & Behavior, Distance Education, Unity College, New Gloucester, ME 04260, USA; kristynrvitale@gmail.com

**Keywords:** *Felis silvestris catus*, cat, free-ranging cat, social behavior, cat colony, social generalist

## Abstract

**Simple Summary:**

Cats are ubiquitous in human spaces. Cats live in our homes and on our streets and occupy a variety of social environments. However, scientists still disagree on the social nature of free-ranging cats (FRCs). This paper aims to review the relevant literature on the social behavior of FRCs and includes which behaviors have been observed and the main findings of each study. The findings of this review indicate that the relationships between FRCs are not random, are socially complex, and deserve further study. The body of literature that currently exists provides an excellent foundation for future work. Further research in this area can help further illuminate the social lives of FRCs.

**Abstract:**

Despite the diversity of social situations in which cats live, the degree to which free-ranging cats (FRCs) are social is still debated. The aim of this review is to explore the literature on the social behavior of FRCs. A search of two major databases revealed that observations of intraspecies and interspecies social interactions have been conducted. The intraspecific social dynamics of FRCs differ based on group of cats surveyed. Some groups display strong social bonds and preferential affiliations, while other groups are more loosely associated and display little to no social interaction. Factors impacting FRC conspecific interactions include cat body size, cat social rank, cat individuality, cat age, relationship to conspecific (kin/familiar), cat sex, level of human caretaking, presence of food, the health of the individual, or sexual status of conspecifics. Interspecies interactions also occur with humans and wildlife. The human’s sex and the weather conditions on the day of interaction have been shown to impact FRC social behavior. Interactions with wildlife were strongly linked to the timing of cat feeding events. These findings support the idea that FRCs are “social generalists” who display flexibility in their social behavior. The social lives of FRCs exist, are complex, and deserve further study.

## 1. Introduction

Domestic cats are ubiquitous across the human landscape. They are found living as companions in human homes and living as free-ranging cats (FRCs) on city streets, in parking lots, on farms, and other outdoor locations where resources such as food, shelter, and mates are plentiful [1]. The social system of FRCs has been described as “facultative sociality” and domestic cats display much flexibility in their social behavior [1,2,3]. FRCs are found living both solitarily and in groups. Cat groups live at low and high densities, ranging from <1 cat per km^2^ to over 2500 cats per km^2^ [4]. Despite the diversity of social situations in which FRCs live, the degree to which FRCs are social is still debated [5].

FRCs include any domestic cat (*Felis silvestris catus*) that has no constraints on their movement. FRCs can be composed of cats of various levels of experience with and dependence on humans. Key terms related to FRCs are defined in Table 1. Individual FRCs will differ in their socialization experience (e.g., duration of socialization, number of social partners, when the socialization occurred in the development of the individual) and the degree to which they are domesticated (e.g., number of generations or strength of selection) [6]. As Udell and Brubaker state, “The interaction between domestication and socialization best predicts the social phenotype an individual will display as an adult” ([6], p. 329).

Social behavior can be defined as any interaction between two or more individuals of the same or different species [17]. These interactions are non-predatory and can be placed into several behavioral categories. These include affiliative behaviors, which are friendly social behaviors that function to form, maintain, or strengthen a social bond. Agonistic behavior is any social behavior related to aggression, submission, or threatening behavior [18]. Reproductive behaviors include any interactions related to mating such as courtship, mounting, and copulation. Caregiving behaviors include social behaviors related to the important survival interactions that occur between a queen and her kittens [19,20].

Although it may seem simple to assign behaviors to specific categories, some behaviors may cross into multiple categories [21]. For example, a social roll (rolling onto back, exposing belly to a social partner) could be considered either affiliative or submissive (see Figure 1a). In these cases, the category of the focal behavior will depend on the context of the behavior. For example, if allogrooming (grooming a social partner) occurs between a queen and her kitten, it would be considered caregiving, but if allogrooming occurs between adult conspecifics, it would be considered affiliative. Additionally some behaviors may initially appear to fall into multiple categories. This can be seen with several behaviors involved in play. Play behaviors may resemble aggression (e.g., biting or cuffing), but they can be distinguished as they are not done to their full extent and they are often preceded or followed by affiliative behavior (Figure 1b, discussed further below). In all, behavioral categories are not always clear-cut but can be clarified by examining the context in which they occurred.

In many animal species, social relationships form between individuals with a history of interactions. In some cases an affiliative bond can form, defined as “long-term relationships established among individuals and characterized by high rates of friendly and peaceful interactions” ([17], p. 1). The idea that FRCs form social relationships has been debated and Spotte stated that FRCs “should be considered solitary—not social” ([5], p. 49). Therefore, the aim of this review is to explore the literature on the social behavior of free-ranging cats. This paper will discuss which social behaviors have been studied in FRCs and the main findings of each study in order to make conclusions about the social nature of FRCs.

## 2. Materials and Methods

### 2.1. Literature Scope

In order to be considered for inclusion in the review, the literature had to be focused on some aspect of the social behavior or social cognition of FRCs (Table 1). Research with pet cats with any constraint on their movement (e.g., indoor–outdoor cats), shelter cats, or cats living in outdoor enclosures were not included. For inclusion, the literature must be original research that directly examines FRC social behavior or an aspect of FRC social cognition (i.e., how FRCs take in, process, and respond to a social partner [22,23]). The research must measure or observe at least one specific social behavior or interaction between a dyad or group of individuals. This includes social interactions with conspecifics as well as non-predatory interactions with other species. Research on human perception or beliefs about FRCs were not considered for inclusion. Work on cat management strategies (TNR or eradication), cat health factors (disease, virus, or parasite transmission), home ranging behavior, activity level, cat abundance, cat density, and studies on cat predatory behavior were not considered for inclusion in the review, unless they measured or observed at least one social behavior. Only published scientific research articles or book chapters were considered for inclusion. These could be an experimental study, observational study, or a case report. Review articles/chapters, commentary, and unpublished theses/dissertations were not included. Only studies published in English were included.

### 2.2. Literature Search and Filtering

The online databases Web of Science and Scopus were used to conduct the literature search. The last date of search for both Web of Science and Scopus was 23 September 2021. The following Boolean phrases were used in the search: (cat OR feli*) AND (roam* OR range* OR feral OR community OR stray OR alley OR colon*) AND (behave* OR social* OR interact* OR cog*). See Table 1 for definitions of the key terms related to FRCs.

This returned a total of 15,087 results from Web of Science and 445,708 from Scopus. The literature was filtered to include only articles (excluding reviews) and to only include results published in English. This returned a total of 13,553 results from Web of Science and 297,620 from Scopus. Results were then sorted by relevance and the first 400 results from both Web of Science and Scopus were taken. A total of 133 duplicate results between the two databases were removed. The titles of 667 relevant articles were read to determine if the paper fit within the scope of the search, this excluded an additional 248 papers. The abstracts of the remaining 419 papers were read to determine if the paper fit within the scope of the search, this excluded an additional 341 papers, leaving 78 papers. After an examination of the paper texts, an additional 49 were removed (e.g., they did not focus on FRCs, they did not measure social behavior directly, study cats were restricted/contained, or the paper was a review instead of research). This left a total of 29 papers, all of which were included in the final analysis. An additional 2 papers were identified after examining references while reading the relevant literature.

## 3. Results

The literature search produced a total of 31 relevant results that were included in the review. Papers were sorted into similar categories. Results indicate the existence of two main areas of research, interactions among FRCs (Intraspecific Interactions, 28 papers) and interactions between FRCs and other species, such as humans and wildlife (Interspecific Interactions, 3 papers). Papers within each area were sorted by behavioral category. In cases where a behavior crossed into multiple categories it was placed in the most relevant category based on the paper’s focus and the context of the behavior, as discussed above. The 31 papers are presented in Appendix A with information on the study location, the behavior(s) measured or observed, and the citation. Photos of social behaviors explored in the literature can be seen in Figure 2 and Figure 3. 

## 4. Discussion

### 4.1. Intraspecific Interactions

The majority of work identified in the search has focused on interactions among cats. In these papers, one or more social behaviors were measured or observed between conspecifics. These papers are reviewed below.

#### 4.1.1. Study Locations

The literature has focused on FRCs living in a diversity of locations. These include cats living at fishing ports, dockyards, and on islands. FRC groups can be highly gregarious in these locations. For example, at the Portsmouth Naval Dockyard in England, approximately 200 FRCs lived in the area [24] and lived at a density of 300 cats per km^2^ [4]. Locations such as dockyards or fishing ports on islands can often support large populations of cats because of the presence of supplemental food. This food can be supplied from people through direct feeding or cats can scavenge discarded food items from ships or trash near the sides of the dock [25,26]. FRCs are especially prominent on some islands, which makes their presence highly controversial. FRC predation can have a detrimental impact on the sensitive ecosystems and endemic species on islands. In these cases, eradication or baiting programs may be common. In other cases, groups of FRCs living on islands are beloved and these “cat islands” can even become popular tourist destinations, as seen in Japan (e.g., Tashirojima, an island in Japan with FRCs seen in Figure 1a, Figure 2b,d and Figure 3). Another “cat island” is Ainoshima, a 125 ha island located approximately 7 km off the coast of Shingu in Fukuoka, Japan. Ainoshima has been the site of several studies on the social behavior of FRCs [27,28,29,30,31].

Groups of FRCs are also common in city centers and residential areas. Research has included FRCs living at locations such as Regent’s Park in London, England [32], in urban and residential areas of Israel [33,34,35,36], and in the urban centers of Rome, Italy. These locations include the work of Natoli and colleagues in the market square “Piazza Vittorio Emanuele” [37], a large courtyard known as “Garbatella” [38,39,40] (which had a density of 2833 cats per km^2^ [41]), and the historical ruins near the center of Rome, “Fori di Traiano” [42]. 

Rural environments are also a common location for FRCs. One of the earliest studies to examine FRC social behavior was conducted by Laundré with cats living on a dairy farm in Green Bay, Wisconsin, USA [43]. One of the best known studies on cat colony social behavior was conducted by Macdonald, Apps, Carr, and Kerby on Church Farm in Bradford, Devon, UK [44]. The colony living on Church Farm was made up of a core group of 4–5 cats with cats from other groups also appearing at the observation site. Finally, two rural populations, Aimargues and Saint Just-Chaleyssin, near Lyon, France, have been the site of several studies on FRC population dynamics and epidemiology [45].

The search identified several locations that range in the approximate number of FRCs observed in the study area. The number of individuals varies greatly, from 4 individuals on Church Farm [44] living at a density of 6 cats per km^2^ [4] to over 200 adult FRCs on Ainoshima [46] at a density of 2350 cats per km^2^ [4]. Intraspecific interactions within these studies were grouped into four major behavioral categories: affiliative behavior, caregiving behavior, aggressive behavior, and reproductive behavior. Results indicate that several factors impact use of these behaviors. Table 2 displays the factor, which behavioral categories are influenced, and summary of the influence.

#### 4.1.2. Affiliative Behavior

Several affiliative behaviors have been observed or measured in research conducted with FRCs. Allogrooming is the behavior of grooming a social partner. During a bout of allogrooming, the cat may lick or gently bite the fur of a conspecific [21]. Allogrooming was observed between FRCs living at the Portsmouth Dockyard [24] and licking was the most frequently noted interactive behavior in the Church Farm colony and made up over half (53.4%) of all interactions observed [44]. Colony cats at Church Farm also displayed allogrooming preferentially. For example, the authors discuss an adult male (only referred to as “TM”) who is the supposed father of two adult females from different litters, referred to as “PI” and “DO”. TM was involved in significantly more interactions with DO and was more likely to interact with DO than with other females (PI or “SM”, another adult female). TM was found to lick DO on 77% of encounters that he initiated and DO was found to lick TM on 40.7% of encounters she initiated. 

Allorubbing is the behavior of rubbing the head or body against a social partner (Figure 2a). Head-rubs were observed between FRCs living at Portsmouth Dockyard [24] and allorubs made up 15.7% of interactions of cats at Church Farm [44]. Additionally, Macdonald et al. found that the presence of allorubbing indicates a social relationship between the cats. Allorubbing can be used as a measure of social position within the colony. Macdonald et al. found that the flow of allorubbing is skewed and tends to be initiated mostly by adult females to adult males, between adult females, from mothers to adult daughters, from adult females to kittens, from kittens to those they nurse on most often, and from previously dominated cats to previously aggressive dominate cats. A linear hierarchy was not seen when looking at allorubbing. However, allorubbing did flow from “peripheral” cats to “central” cats found in the colony core.

Spending time in close proximity or engaging in bodily contact, such as sleeping together, are also affiliative behaviors noted in FRCs (Figure 2c,d). In the Church Farm colony, cats tended to spend time near one another [44]. All adults spent time near another cat (within 10 m) in over 50% of observation scans. Cats also preferred different sleeping arrangements. Some individuals preferred to sleep alone while others slept in bodily contact with conspecifics (Figure 2c). Cats were found sleeping alone on 31.7% of scans and found in bodily contact with one or more cats on 16% of scans. However, individually, one cat was found sleeping alone on 70.9% of scans while another cat was found sleeping alone on only 32.1% of scans. Instead, this cat spent time sleeping near one other conspecific on 43.1% of scans and with multiple conspecifics on 24.9% of scans.

In another study, Page and colleagues examined the home ranging behavior of FRCs living at Avonmouth Dockyard in Bristol, England using radio tracking [26]. Observations included notes on “actual contact”, or instances in which two cats were aware of each other’s presence. Although cats showed substantial overlap in their home ranges, they often did not engage in “actual contact” with one another. Sixty-three percent (120/190) of ranges overlapped with one another; however, only in 33 of these overlapping cases was “actual contact” observed between the cats. This may indicate that although these cats share the dockyard location spatially, they may not be sharing the location temporally. However, individual variability was noted. Some males were regularly noted to spend time together and one male spent a majority of his time with a specific female.

Social play, or when two or more individuals engage in play with physical contact or toward a mutual object, has also been observed in FRCs (Figure 1b and Figure 3c,d). Social play between conspecifics can involve behaviors such as chasing, cuffing (slapping a conspecific with the paw), and wrestling. During a bout of wrestling, cats may bite one another (especially on the front or back of the conspecific’s neck) and rake their back legs against the conspecific (i.e., rapidly move their legs back and forth) (K. Vitale, Observation). As mentioned, behaviors involved in play may overlap with those categorized as aggression (e.g., chasing, cuffing, biting); however, they are distinguished from aggression in that they are conducted in a “non-serious” manner and the cat shows no intent to harm [21]. Play can be identified from a lack of aggressive vocalizations (e.g., growling, caterwaul), brief pauses in the bout in which cats are distracted (i.e., attention shifted away from conspecific to other aspects of the environment before re-initiation), behaviors are inhibited (i.e., bites do not draw blood, cuffs are done with claws retracted), and play may be preceded or followed by other affiliative behavior (K. Vitale, Observation).

For cats in the Church Farm colony, social play was observed in some dyads frequently and rarely or never in other dyads [44]. Macdonald et al. report the percentage of initiations by one cat to another cat for each behavior. Although TM (adult male) initiated licking with SM (adult female) for 70.1% of interactions he initiated, TM never initiated play with SM. TM only initiated play with one cat, PI (an adult female), but even then it was rare (only 0.6% of initiations). However, TM was a frequent receiver of play from a kitten, “LU”. For the encounters LU initiated with TM, LU initiated play on 50% of instances. In fact, LU engaged in much social play. With the adult females, the percentages of all LU’s initiations that were play bouts were 42.6% with SM (LU’s mother), 65.9% PI, and 83.9% DO. That is not to say no adult females initiated play. It was found that 29.3% of PI’s initiations with DO were play and 23.4% of DO’s initiations with PI were play. However PI and DO initiated play with other conspecifics less (between 8.4–13.4% of initiations). This work indicates that social play between FRCs varies individually and can be preferential toward some conspecifics and not others.

Affiliative behaviors are also common during greeting events. Two behaviors related to greetings between FRCs include the nose sniff, in which two cats approach one another and touch noses, and the “tail up” signal, in which a cat sticks its tail into an upright position. The tail tip may bend slightly to one side [21,39], (Figure 2b). At the Portsmouth Dockyard, the most commonly observed greeting behavior was the tail up signal [24]. A study conducted at the Garbatella in Rome examined the social function of tail up [39]. It was found that tail up was often seen alone and was not often paired with additional social behaviors, like allorubbing or nose sniffing. Tail up and another affiliative behavior co-occurred on only 22.7% of instances. However, cats did show individual variation. Some cats frequently displayed tail up prior to allorubbing and also tended to display the behavior preferentially toward specific conspecifics. Nose sniffing was rarely seen to co-occur with tail up. In terms of social rank, a cat’s position within the group impacted use of tail up. Lower ranked cats displayed the tail up more frequently and higher ranked cats received the tail up more often. These results indicate that social rank within the group impacts use of the tail up signal [39] with it being used more commonly by lower ranking individuals toward higher ranking. In all, Cafazzo and Natoli state that “Tail up is not simply a greeting behavior” (p. 64).

Results from these studies have highlighted several factors that impact use of affiliative behavior (Table 2). The cat’s sex may impact the use of affiliative behavior and the social partner to which the behavior is directed. In several studies, adult females tend to be the initiators. At Portsmouth Dockyard, it was observed that affiliative behavior was initiated more frequently by female cats than by toms and toms were not noted to engage in affiliative behavior with one another [24]. Female cats also displayed more affiliative behavior toward more familiar males. In the Church Farm colony, males were typically the recipient of interactions from females and the females the recipient from kittens [44]. At Regent’s Park, affiliative behaviors were often seen between females and mature males or among females, but males did not display this behavior toward one another [32]. At Garbatella, it was found that females typically displayed tail up and allorubbing towards males; however, males often displayed nose sniffing towards females [39]. On the other hand, for the Avonmouth Dockyard cats, no “actual contacts” between females were observed but some males were regularly seen together [26]. Laundré also noted that males initiated more affiliative encounters than females and the majority of these affiliative encounters were due to the actions of two individual males [43]. Females in this group also preferred female–male affiliative encounters to female–female. Although sex appears to be an important factor impacting use of affiliative behavior, the exact influence may vary due to the individuality of the cats involved, the dyad’s relationship, or environmental conditions (such as cat density, food availability, or food distribution).

Spaying or neutering also appears to be an important factor impacting affiliative behavior. In one study, affiliative behaviors became more common after neutering for both sexes [32]. Another study, which observed 184 individuals in 4 feeding groups (2 groups of which were neutered and 2 which were intact), found that compared to the neutered groups, one of the unneutered groups had a significantly higher rate of affiliative behavior per cat [33]. Another study at Garbatella found that, following neutering, cats spent less time in close proximity to conspecifics [40]. However, some male–male dyads engaged in increased allorubbing and nose sniffing. Males who had never engaged in these affiliative behaviors prior to neutering did so following neutering, and these males also spent slightly more time in proximity to each other after neutering.

As mentioned, the individuality of the cat itself may impact use of affiliative behavior. In the Church Farm colony, some cats tended to initiate interactions whereas other cats tended to be the receivers of interactions [44]. Cats also displayed “preferential affiliations” or individual preferences for certain conspecifics within the colony. Macdonald and colleagues stated about the Church Farm colony, “there are many non-random relationships between these cats” (p. 45). On the other hand, Laundré noted few affiliative behaviors between cats on the dairy farm [43]. Those behaviors that were observed included allorubs and nose touches that occurred during greetings and, as mentioned, were primarily due to the interaction of two individuals. The body of work examining FRC affiliative behavior indicates that there is variability in both the sociability of individuals in the group as well as the associations between group members.

#### 4.1.3. Caregiving Behavior

Caregiving behaviors are important for the survival of the kittens and in forming an attachment bond between a queen and her kittens. These include attachment behaviors, such as proximity and contact seeking, as well as nursing, denning behavior, and allogrooming [19,20]. In one study, conducted on Ainoshima, Izawa and Ono examined relationships between queens and their offspring [30]. Around 200 adult cats lived in this area and 72 breeding sites with litters were observed. Females were observed engaging in affiliative behavior, such as resting together. Some of these females were related pairs, such as mother–daughter dyads, and others were females with unknown relationships. Both males and females showed tolerance toward kittens, but only females displayed care toward kittens. Cooperative nursing was observed; females were noted to groom and suckle kittens that were not their own offspring. Two mothers were seen to share in caregiving of kittens without distinguishing their own. Throughout the study, 19 instances of cooperative care were observed, and, of these, 9 were observed in mother–daughter pairs, 5 were observed between littermates, and in the remaining 5, the kinship off the dyads was unknown.

The caregiving behavior of cats living on the Church Farm was also observed [44]. Communal denning and alloparental behavior were also observed exclusively in females and made up 7.6% of observed cat activity. Adult females were also found to nurse and care for kittens indiscriminately. In 2 years of observation, all 12 breeding female dyads that could have nursed each other’s offspring were observed doing so and 8 out of 9 dyads were observed sharing a nest with kittens. For adult female–kitten dyads, most initiations by females to kittens were of the adult licking the kitten. On the other hand, kittens initiated more allorubs than adult cats. However, the number of initiations by kittens to females declined as the kitten aged. Kittens behaved discriminately between females in both the frequency and quality of their interactions, which indicates the presence of social relationships. However, they also nursed from adult females other than their mother, indicating cooperative care.

#### 4.1.4. Agonistic Behavior

Research has examined both aggressive and submissive behaviors in FRCs. Aggressive behaviors include engaging in aggressive posturing (also charging or chasing), aggressive vocalizations (e.g., growling, caterwauling), and cuffing. Submissive behaviors include fleeing (quickly leaving location), social roll, and exhibiting a small/crouched body posture [21].

Cats may also engage in “actual fighting”, or aggressive encounters involving direct physical contact. However, at the Portsmouth Dockyard, few fights involving direct physical contact between cats were noted [24]. Instead, the majority of agonistic interactions involved ritualized threat postures and vocalizations. One behavior of note was head aversion. Instead of looking directly at one another, cats averted their heads at right angles to avoid a direct stare. Although fights were uncommon, male aggression towards females was especially rare.

Similarly, aggressive behavior on the Church Farm was infrequent [44]. Aggression was rarely observed between adult members of the core colony and only made up 4.9% of observed interactions. However, during encounters between colony members and cats of an outside group, aggressive behaviors were common and made up 53.7% of intergroup interactions. In these cases, cats in the core colony almost always served as the initiator of the aggression. Often the initiation of aggression against an outside cat by a core cat would result in the cooperative behavior of additional group members to chase off the stranger.

Social rank can also be examined through the outcomes of agonistic interactions. For example the social rank of colony cats living at Garbatella was examined [38]. Each individual within a dyad was ranked. Cats classified as “dominant” were those who engaged in more aggressive behavior than was displayed to them or received more submissive behavior than they displayed. Cats of higher social rank displayed more aggressive behavior; the higher the rank, the more aggressive the cat. Laundré described a female social hierarchy with an “alpha” and “beta” female [43]. However, social rank was not well documented below the dominant females and male ranking was not determined because they displayed little aggression toward one another. Finkler and Terkel found that intact females who were more socially dominant also had higher cortisol levels (a common stress measure, high cortisol indicates higher stress) compared to intact females who were less dominant [36].

Given that FRC groups tend to form around locations of plentiful food, it is unsurprising that the presence of food is also a key factor impacting social behavior. In general, Laundré noted aggressive encounters were infrequent among cats [43]. The exception to this was near feeding time and almost all (97.5%) aggressive encounters occurred when milk was given to the cats. As the time to feeding grew closer, cats would gather in the feeding area and an increase in aggression was noted. A sharp increase in aggression was observed 1 min prior to feeding, ceased at the moment of feeding, and increased again about 30 s following food delivery until finally dropping again after 5 min, when individuals had begun to disperse. This is supported by another study, which found the gathering of cats around the feeding site was accompanied by an increase of agonistic behaviors [33].

The presence of food also impacted social rank. Males were found to occupy more dominant positions in the absence of food while females increased in social rank at the feeding site [38]. Although kittens were ranked at the bottom of the hierarchy, they were often the first at the site to feed. This, along with the findings of Izawa and Ono that kittens were tolerated at their mother’s feeding group, and that adults of both sexes showed infrequent aggression toward kittens [30], indicate that adult FRCs show a level of social tolerance for kittens.

It is possible sex or alterations to sexual activity are also important factors impacting aggressive behavior. As mentioned, male aggression towards females can be rare [24]. However, the use of aggression does not appear to differ between sexes. For cats living at Garbatella, aggressive behavior did not significantly differ between males and females [38]. In terms of sexual status, the impact of neutering on aggressive behavior is mixed. In one study, aggression was rare both before and following neutering [32]. In other work, neutered groups of cats displayed less aggression than unneutered groups of cats [33] and compared to intact cats, neutered cats showed lower frequencies of agonistic behavior [35]. Aggressive behaviors have also been compared with cortisol [34]. Compared to intact females, neutered females displayed less aggression and reduced cortisol levels. Intact females who displayed more aggression also had higher cortisol levels compared to the less aggressive intact females.

Other factors may also be important. Feeding context can impact the effect of neutering. In the absence of food, aggressive behavior decreased following neutering [40]. However, neutering had no effect on aggressive and submissive behaviors in the presence of food. In terms of social rank, cats remained stable over time and the hierarchy order did not change after neutering. Finally, level of human caretaking may also impact aggressive behavior. In one study, cats who had minimal human care displayed higher aggression than cats who received maximum care [35]. In all, human activities such as sterilization, provisioning of food, and level of care, as well as traits of conspecifics (such as their age or relationship to one another) are all important factors impacting aggressive behavior to varying degrees (Table 2).

#### 4.1.5. Reproductive Behavior

Reproductive behaviors, such as courtship, mounting, and copulation, have been observed in FRCs. For cats at the Portsmouth Dockyard, reproductive behavior was noted to occur in most months [24]. The most frequently observed courtship behavior was waiting. Waiting involved the male staying near to the female and following her if she walked away. During mating, FRCs produce a variety of vocalizations. Males and estrus females at the Avonmouth Dockyard would vocalize to one another when walking separately [26] and male cats on Ainoshima produced rutting vocalizations in addition to behaviors related to courtship and copulation [27,28]. Males may also engage in vocal dueling [47].

In general, reports of aggression between males during courtship were infrequent. There was no evidence of competition between males for access to estrous females [26] and no fighting [24] or aggression was noted between waiting males [47]. When aggressive interactions were noted during courtship, they were observed between toms and females and were initiated by unreceptive females [24]. An exception to this is on Ainoshima, where aggressive encounters were frequent when males gathered around a female after she entered estrous [27,28,29,31].

Several factors may impact reproductive behavior, including body size, membership to the social group, and cat individuality. Kinship is also an important consideration. It was found that females avoided copulation with their kin, at least closely related kin [31]. Males also made less mating attempts with related females and any attempts that were observed were less frequently accepted by the females. In terms of body size, heavier males tended to court more females [28] and an individual male’s body weight was significantly correlated to the number of copulations as well as the copulation rate. This indicates that heavier males have higher mating success [27]. In another study, males with larger body size had a higher social rank within the group and also had higher mating success [29]. However, not all work has consistently shown a relationship between body size and aggression. Another study found no correlation between average copulation success or mounting success and male body weight or male age [31]. However, there was a relationship with female body size and acceptance of male attempts. Compared to heavier females with longer cycles, females who were lighter and had shorter estrous cycles accepted mounts more frequently. Additionally, in comparison to younger and heavier females, older and lighter females also accepted copulation more frequently.

Another factor impacting reproductive behavior is whether the conspecific is a member of the same social group. In one study, some males would court and copulate only within their own group and did not copulate with females of other groups. Interestingly, this behavior to stay within the group was often noted in lighter males. On the other hand, heavier and mid-sized cats would court females of outside groups in addition to females of their own group [28]. However, when males did attempt to court females of other groups, they were often unable to approach these females as closely as they could females of their own group. Even when these males did attempt to mate with females of other groups, their attempts were often defeated due to the aggression of lighter males from within that social group. This type of behavior reduces the success for intergroup copulations [27]. The focus to stay within the group was also seen for female copulations. Females copulated more often with males in their own social group. Seventy-seven percent of copulations occurred with males of the same social group, 12.9% with males of a different group, and 9.7% with males who had no group membership [27]. This indicates that membership to a social group impacts FRC reproductive behavior.

The individuality of the cat can also impact reproductive behavior. On Ainoshima, each male courted approximately 3 females but there was individual variation and studies ranged from 1–11 females [27] to 1–9 females [28]. Additionally, some males would court a female for a long duration of time (>10% of observations) while others would only court for a short period of time (<5% of observations) [37]. In another study, some males spent more time courting females or courted more females than other cats [47]. Females also displayed individual variation in receptivity to male mounts and copulation; however, females tended to be choosier with males during copulation than during mounting [31]. Additional observations at Garbatella, Fori di Traiano, and the Croix Rousse Hospital Park indicate that cat individuality is linked with annual reproductive success. Males with bold temperament were found to have the highest reproductive success [41].

An individual’s health status may also impact reproductive behavior. A bold temperament has also been linked with a higher probability of infection with Feline Immunodeficiency Virus (FIV). An FIV+ cat has a disease caused by the virus that compromises the cat’s immune system. Another study in the Croix Rousse colony explored whether FIV impacted FRC mating behavior and found males infected with FIV mounted females the most [48]. Of the seven mountings noted, five were done by males infected with FIV. Additionally, socially dominant males tended to be infected by FIV [49]. This makes sense as FIV is transmitted through bites, which may occur during actual fights. This may indicate FIV transmission is impacted by the social status of the cat and occurs most often by socially dominant males.

Social rank may also impact other reproductive behaviors. In one study, there was no difference in time spent courting females based off social rank [49]. Male cats that were higher ranking did not invest significantly more time engaging in courting than lower ranking males. However, another study at Fori di Traiano analyzed the directionality of aggressive and submissive behavior as a measure of social rank. Here, social rank was found to be a factor impacting male reproductive success. The most dominant male sired the highest percentage of kittens. However, cats of higher rank did not control access to females and 78% of litters had multiple paternities.

Additional social behaviors may occur, such as the male–male (MM) mounting observed in FRCs on Ainoshima [29]. Males were either those who mounted other males (MM males), males who were mounted by other males (RM males), or males who showed no male mounting behavior (NM males). In one year, a total of 26 cases of MM mounting were seen from 14 male dyads. For 50% of these encounters the behavior of thrusting was also seen; however, it was unknown if ejaculation occurred. Of these 26 instances, 20 occurred when males had just switched their attention away from a female to a nearby male and 6 cases occurred after the estrous female ran away. Age impacted MM behavior. Older males tended to mount while younger males tended to be mounted. This indicates MM mountings were likely not “practice” for females (as more MM would be expected in younger males). Contrary to the findings between males and females in terms of staying within the group [27,28], the majority of MM mountings occurred between males of different groups (19 MM intergroup and only 7 MM intragroup) [29]. MM mounting was only ever observed inside of female estrous seasons. The presence of MM behavior did not impact female-directed behavior and MM and NM males had similar success in courtship and copulation.

One final note should be made on the documented instances of infanticide in FRCs. Six spontaneous occurrences of infanticide were observed directly by Pontier [45]. These occurrences were noted while collecting data for another other research project. The number of kittens killed ranged from 1–4 with an average of 3. All of the infanticidal males appeared to be adults, were unknown to the author, and did not belong to the focal group. All kittens that were killed were young, within the first weeks of life, and were killed from the male cat biting their necks, and then holding and vigorously shaking them. This killed the kittens immediately and following this, the male would leave the kitten intact. Only one instance differed from this behavior. In this case the kitten was killed from an abdominal bite and the body was partially eaten. In response to the infanticidal behavior, all females were observed to engage in aggressive behavior toward the males. In one case, an infanticidal male had killed a kitten, but was then driven off by both the female (who had previously lost kittens in another infanticide case) and a resident male. Although the cooperative behavior was successful, both male and female residents were injured from the interaction.

Because this population had been monitored for several years, the level of infanticide was known to be relatively rare, occurring for only 6.6% of litters [45]. An instance of infanticide was also noted at Church Farm [44]. One unfamiliar male, who was not a member of the colony, was noted to have killed six kittens (by biting the back of the skull) before three adult females attacked and chased off the male [44]. As discussed by Pontier and Natoli, in general, this behavior appears to be relatively rare for cats in rural environments and absent for cats living in urban environments [45]. One possibility for this is a difference in terms of accessibility of mates. Male cats living in urban environments, especially highly dense environments, have more opportunity to reproduce with females while males in rural environments, where mates may be more dispersed, have less opportunity to reproduce and may therefore need to engage in more competitive behavior for access to females. However, it is also possible this behavior is atypical, due to other environmental conditions, or has a genetic basis. In all, the authors suggest that kitten infanticide appears to be more of a “behavioral tactic” rather than a “specific feature of cat population biology” (p. 448).

### 4.2. Interspecific Interactions

A total of three papers were identified through the search as focused on interactions between cats and another species. Two papers focused on interactions between FRCs and humans and one study focused on non-predatory interactions between FRCs and wildlife. 

#### 4.2.1. Study Locations

FRCs can be found living near areas of human activity, where resources are available and there is ample opportunity for social interaction. For example, in Japan, FRCs can be found living on islands, near shrines, and on city streets. FRCs often approach people in these locations, rub on them, and accept social interaction such as playing and petting (Figure 3). One identified study was conducted on the campus of Nanjing University in the Jiangsu Province of China [50]. College campuses are a common location for FRCs and cats are often provided food at feeding stations. Rural locations, next to open fields or woods, can also be the site of FRC groups. One case study identified in the search examined a FRC group living outside of a mental health center in Alma, MI, USA [51]. Finally, FRCs live in a variety of environments and in conditions that may attract other wildlife, such as locations with feeding stations or waste sites. Only one paper was identified which examined FRC interactions with wildlife that occupy the same habitat as cats. The FRCs lived on Jekyll Island, a 5847-acre barrier island on Georgia’s Southeastern coast [52].

Interspecific interactions were grouped into two major behavioral categories: affiliative behavior with humans and non-predatory interactions with wildlife. Results indicate that several factors impact use of these behaviors. Table 3 displays the factor, which behavioral categories are influenced, and a summary of the influence.

#### 4.2.2. Affiliative Behavior

Affiliative behaviors occur between FRCs and humans. These may include seeking to be near to or in contact with humans (such as accepting petting or sitting on a person’s lap) and engaging in play with people (Figure 3). In the study of FRCs living on Nanjing University campus, cats were assessed for their response to an unfamiliar person [50]. Specific affiliative behaviors were noted and each social behavior was given a score. Affiliative behaviors included approaching the person, spending time in proximity to the person (within 1 m), allorubbing (on legs), vocalizations (such as meowing and purring), and social play. The cat’s response to petting and holding were also noted. The sum of all the social behaviors shown toward the unfamiliar person was each cat’s Socialization Score. A high score indicated high sociability toward people and a low score indicated low sociability. Results indicate that FRCs display affiliative social behavior toward humans. Factors impacting these behaviors include the person’s sex and the weather conditions at the time of interaction. Not considering the weather, cats were more sociable and friendlier to women compared to men. Not considering the person’s sex, cats were more sociable toward people on days with bad weather conditions (rain or snow present) compared to good weather conditions (no rain/snow). Considering the interaction between human sex and weather, cats were much friendlier to women on days with bad weather.

FRCs have also engaged in social behavior with humans in unexpected ways. The use of Animal-Assisted Interventions (AAI), or interventions which incorporate an animal into the therapeutic process, have become more commonplace. Dogs are commonly used in AAIs. However, the use of non-traditional animals, such as cats, has received less attention. Yet, in a 1997 case study reported by Wells and colleagues, four FRCs unexpectantly became involved in therapy sessions [51].

The process by which these FRCs became integrated into therapy sessions started when staff members of the health center started providing supplemental food to the cats. The cats lived directly outside of the office building and individuals in the therapist’s office were able to clearly see the group of cats. The authors reported that many of the clients stated they looked forward to seeing the cats on their visits and clients had asked if the cats could be allowed into the office. At one point, a window was left open in case a cat wanted to enter. Eventually a mother and her two kittens entered through the window. Staff and clients never restricted the cats’ movements or attempted to control their behavior. However, if the cat initiated interaction or play, the client was allowed to reciprocate the interaction. All three cats eventually became familiar with the office space and spent time in preferred locations where they would sleep and perch. Over time, the cats became more comfortable around people and the authors reported that eventually all cats accepted physical touch from humans. By this point, the cats were a staple in the office and had become integrated into the therapeutic process.

Although social behaviors themselves were not specifically analyzed (e.g., using duration or frequency measures), social behaviors were reported between FRCs and clients. Affiliative social behaviors initiated by one or more of the FRCs include approaching and greeting people and initiating physical contact with people (e.g., tactile contact, sitting on the client’s lap). In terms of client behavior toward cats, many clients vocalized to the cats and called the cats to them. Some clients appeared to become attached to the cats and would pay special attention them. These clients would frequently ask that the cats be allowed in at each appointment and showed interest to learn the cats’ names and histories.

Wells and colleagues report that FRCs have the potential to be “co-therapists” who can assist patients with physical or emotional trauma and that the presence of FRCs allows patients to take an important step toward recovery [51]. In this case, FRCs allowed patients to feel safe and valued (e.g., if a cat chooses to come in the window or engage in affiliative behavior), allowed clients with physical trauma to practice tactile interaction with a non-threating social partner, allowed people with attachment problems to establish and keep a healthy attachment bond with a social partner, allowed clients to develop empathy, and benefited the discussions between patient and therapist. In all “therapeutic themes” began to emerge during sessions with these cats. In many ways, the cats served as a symbol of resilience that mirrored the experience of the clients. It was not just any population of cat, but FRCs specifically, which benefited the patients.

#### 4.2.3. Non-Predatory Interactions

More general interactions have been examined between FRCs and wildlife. On Jekyll Island, 31 FRCs had a KittyCam placed around their neck to record possible interactions with wildlife. Researchers compared the interactions of cats during feeding periods (2 h before/after the provisioning of supplemental food) to interactions outside of the 4 h period. Encounters between cats and wildlife were only counted if the cat displayed behavior that indicated they were aware of the presence of the other animal (e.g., cat changes head positioning to look up at the other animal), and if the interaction was between a cat and an animal of similar or larger size to cats.

A total of 142 interactions were recorded between cats and wildlife [52]. Duration of the interactions between cats and wildlife ranged from a few seconds to more than 30 min. Of these interactions, 49% were with raccoons, 37% with black vultures, 13% with white-tailed deer, and 1% with Virginia opossums. Interactions with all animal species occurred at a mean time of 09:39 h and were linked to the timing of the feeding period. A significant majority of these interactions occurred within two hours of feeding. Although many interactions did not involve bodily contact, at times, cats were observed hissing and batting at raccoons. Cats could also be found in close proximity of wildlife (<30 cm), which may pose an issue for transmission of pathogens between these populations. For FRCs at Avonmouth Docks, three interactions between foxes and the FRCs were noted [26]. The authors only mention that this interaction did not involve physical contact but do not provide further details so it is unclear what these fox–cat interactions entailed. Although stray dogs were observed in the dockyard area, no interactions were seen between free-ranging dogs (FRD) and FRCs. In all, this work indicates several animal species are attracted to locations with supplemental feeding and FRCs interact with some of these wildlife species in non-predatory ways.

### 4.3. Future Directions

The body of work presented in this review provides an excellent foundation for future work in the area of FRC social behavior. The majority of the identified literature focused on intraspecific interactions. Two behaviors that are observed in FRCs, but have not received much attention, are the social roll and social play (Figure 1). Future research should explore if FRCs display play signals (similar to the play bow in dogs) and what factors influence use of a social roll. Additionally, only two papers in the area of FRC–human interaction have been published to date. Despite the fact that FRCs are ubiquitous in human spaces and that humans engage in various levels of FRC caretaking (e.g., provisioning supplemental food and shelter, TNR, medical care, etc.) there remains a lack of work in this area. Key questions for future work include: Does human behavior impact the behavior of FRCs (i.e., does human presence, vocalization, or touch impact social interactions between cats)? Do FRCs form bonds with humans? 

Additionally, no research was identified that examined aggressive behavior between FRCs and humans. Although, aggressive behaviors were not directly measured in the case study of FRCs in the therapy office, no aggressive interactions in which clients were harmed were reported [51]. Clients were reminded that cats can be unpredictable and this may have regulated the behavior of the clients to some extent. Additionally, the level of human caretaking was found to impact cat–cat aggression (Table 2). In all, there are likely many ways that human behavior impacts the social behavior of FRCs toward people and other cats. Future work should explore these complex social interactions. Possible research questions include: Does the presence of food also impact aggression of FRCs toward people? Does socialization to humans impact cat–cat aggression?

There are also some considerations to keep in mind when reading the present review. Several relevant studies, some of which were conducted in the same study locations described in this paper, were identified in the search. However, they were not described here because they fell outside of the scope of the paper. For example, in one study, Yamane and colleagues examined cat feeding behavior on Ainoshima but the authors did not measure social behavior directly. However, the authors did note tolerance toward kittens in terms of feeding behavior [53]. Turner and Mertens also observed cats living on three Swiss Farms. However, they measured social tolerance through overlap in home ranges and not through direct social behavior. Turner and Mertens found “The general pattern of social organization found elsewhere was confirmed: males were generally more tolerant of each other than females (based on range overlap), especially considering animals living on different farms.” [2]. Finally, because this review focused only on social behavior, some aspects of studies included in this review are not discussed. For example, Page et al. also measured additional behaviors of individual cats, such as home range behavior and scent marking behavior [26]. Future reviews can draw on a wider range of studies to consider if factors such as the density in which cats live (high vs. low population density) and an individual’s feeding or home ranging behavior impact social dynamics between cats. Finally, only studies published in English were included in this review and future work should expand to search additional languages. Given much of this work has been conducted in countries where English is not the primary language (e.g., Japan, Italy) it is possible additional work has been published that was not captured within the current review.

No research was identified within the area of cat social cognition that directly measured an interaction between social partners. As outlined in the scope of this review, only research that measured or observed at least one specific social behavior or interaction between a dyad or group of individuals were included. However, behavioral studies in which other aspects related to social cognition do exist and were captured in the initial search, such as olfactory perception of conspecific odors (e.g., [54,55]). One of these studies explored the ability of cats to discriminate social partners from odor alone. Natoli found that cats investigated urine sprayed from an unfamiliar male tom longer than urine sprayed from a familiar individual. This indicates that FRCs discriminated between familiar and unfamiliar odors [54], similar to results seen with pet house cats [56]. Given the variability of cat sociability toward humans, with some cats being highly social [57], the fact FRCs display social behavior to people, the prevalence of FRCs in human spaces, and the success of researchers currently studying socio-cognitive abilities in FRDs [58,59], it appears research into FRC–human social cognition is an important area of future research. Key questions include: Do FRCs alter their social behavior in response to human emotional state or attentional state? Do FRCs respond to human gestures (e.g., gaze and pointing)? Do these cognitive abilities vary by the socialization history of the FRC?

## 5. Conclusions: Solitary or Social?

A review of the literature on FRC social behavior highlights several key findings. The first is that several factors influence FRC social behavior (Table 2 and Table 3). For conspecific interactions, these factors relate to characteristics of the cat’s themselves (e.g., sex, age, sexual status, and body size), relationships to the conspecific (e.g., kin, familiar individual, and group member), or factors in their environment (e.g., level of caretaking). For interactions with humans, factors in their environment, including the sex of the human and weather conditions, have been found to impact FRC–human social behavior.

Additionally, FRC groups display much variability in their social behavior. The intraspecific social dynamics of FRCs differ based on group of cats surveyed. Some cat groups display strong social bonds with preferential affiliations among group members (e.g., [44]) while other cat groups are more loosely associated and display little to no social interaction [60]. Even FRCs living in similar environments can differ greatly in terms of social behavior, as seen in the studies with dockyard cats. Dards reported that Portsmouth FRCs lived socially in groups [24] while Page et al. found that Avonmouth Dockyard cats displayed primarily solitary living [26]. In some cases, a collection of cats may not qualify as a social colony, especially for groups that primarily engage in solitary behavior or only have loose affiliations (See Table 1). However, it is evident from a review of the literature that cat groups often have non-random associations and social relationships within the colony.

For cats that do form social groups, the use of affiliative behavior is often seen. Familiarity between cats seems to be an important factor impacting use of affiliative behavior. Familiarity between conspecifics may also be one important factor impacting the formation of “preferred associates” [24,44] and kinship may play a role as well [44]. Cooperative behaviors between FRCs were also noted, specifically among members of the same social group. These include behaviors such as communal denning, and cooperative care in which adult females nursed and cared for kittens indiscriminately [30,44]. Additionally, physical fights involving direct contact between cats are infrequent. Instead, agonistic interactions that involve submissive behaviors and body posturing (e.g., use of head aversion to avoid eye contact) are observed. It is possible that a lack of social relationships within a group may increase aggression between cats in the same location [43].

In all, Dards stated of the Portsmouth Dockyard cats, “It is evident that the domestic cat is capable of a much more complex social structure than has previously been thought, and that this also extends to the adult males, which at first sight appear to be independent of it” ([24], p. 151). Similarly, Macdonald et al. state of the Church Farm cats, “there are many non-random relationships between these cats” ([44], p. 45) and these findings, “indicate that the colony had a social structure and was not simply an aggregation and that the cats’ social interactions were structured according to distinct social relationships” (p. 59).

Although several of the reviewed studies have found that groups of cats form colonies with distinct social relationships, not all studies of cat social behavior have found the same results. Page et al. note of the Avonmouth Dockyard cats, “The cats were mostly solitary rather than group-living, with little contact or social interaction” ([26], p. 263). Additionally, cats at a waste site were often observed eating rubbish simultaneously, but cats remained mainly separated and interaction was infrequent [60]. Cats were observed interacting on only 35 occasions out of 330 observations, which accounts for just 10.6% of observations. These examples highlight the social flexibility of the domestic cat and its ability to form strong bonds as well as loose associations.

This review explored the work to date that has examined the social behavior of FRCs in order to make conclusions about the social nature of FRCs. Spotte stated, “That the domestic cat is social at all, much less socially complex, is doubtful” [5]. Although it is true that “Increased tolerance of conspecifics is not necessarily a sign of willing social interaction” (p. 50) and that “aggregation itself is not evidence of sociality” (p. 70), the review of the present literature indicates that in several instances, the social behaviors of FRCs extend beyond that of mere tolerance. Although groupings of some cats may be more accurately described as displaying “mutual tolerance” with loose affiliations [42], this description does not accurately fit many of the behaviors exchanged between cats in these colonies or during interactions with people. FRCs choose to initiate social interactions with humans and conspecifics and can be highly social. As mentioned, an affiliative bond is an enduring social relationship noted by high rates of affiliative behavior [17]. Cats do not display social behavior randomly. In many cases, cats within the colony appear to display affiliative bonds with certain cats over others, the direction of these interactions appears to be stable, and cats from outside groups are likely to receive increased aggression from group members. These findings indicate that the relationships between FRCs are indeed complex and deserve further study.

Domestic cats are not a unique case in this display of flexible social behavior. Variability in the social behavior of the domestic dog has also been noted [6]. Domestic dogs also live in a variety of social environments, both in human homes and as FRDs outdoors. Domestic dogs also have a sensitive period for socialization (~3–12 or ~3–16 weeks of age) and similar to cats, their social behavior is dependent on the experiences they encounter during this time. Dogs not socialized during this window would also be considered “feral” and dogs that were socialized during this window would be considered “tame” (see Table 1 and [6] for a discussion on FRDs). Some scientists have also described domestic dogs as “facultatively social” [6,61]. The social living of FRDs is also known to vary depending on environmental factors such as resource availability. This variation may be due to a “social plasticity” that allows dogs to occupy a range of social environments. Udell and Brubaker define a social generalist as a “species that can thrive in many different settings as a result of an ability to adapt to a wide variety of social environments and adopt different social strategies “(p. 327).

As a species: domestic cats can also be considered as “social generalists” that display flexibility in their social behavior. Research with housecats indicates that some pet cats become stressed from living without conspecifics while other cats are stressed by the presence of conspecifics [62]. Pet cats also show variation in their sociability toward people and many cats display the capacity to be highly social [57]. Pet cats have also been found to form strong bonds to humans [63,64,65] and these relationships are relatively stable over time [63]. Finally, many cats seek out social interaction from people [57] and many cats prefer social interaction with humans, even over other preferred rewards like food, toys, or scent objects [66]. In all, the findings of this review suggest that FRC social behavior is highly individual, socially flexible, and dependent on a number of lifetime and genetic factors. The social lives of FRCs exist and they are complex. The present body of literature provides an excellent foundation for future work. Continued work in this field is important and will help further illuminate the social lives of FRCs.

## Figures and Tables

**Figure 1 animals-12-00126-f001:**
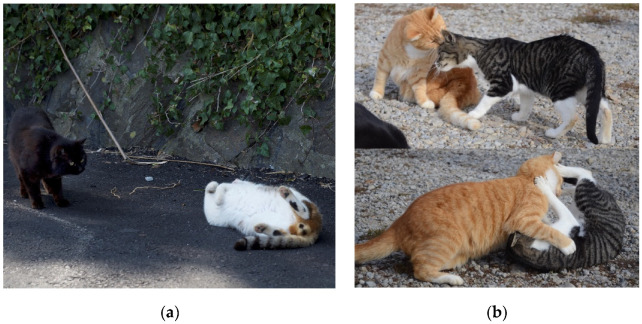
Photos of FRC social interactions taken by K. Vitale: (**a**) a social roll is displayed from one FRC to another; (**b top**) a male tabby cat rubs his head against an orange male as a greeting; (**b bottom**) interaction continues into play and full contact social play is seen between the dyad.

**Figure 2 animals-12-00126-f002:**
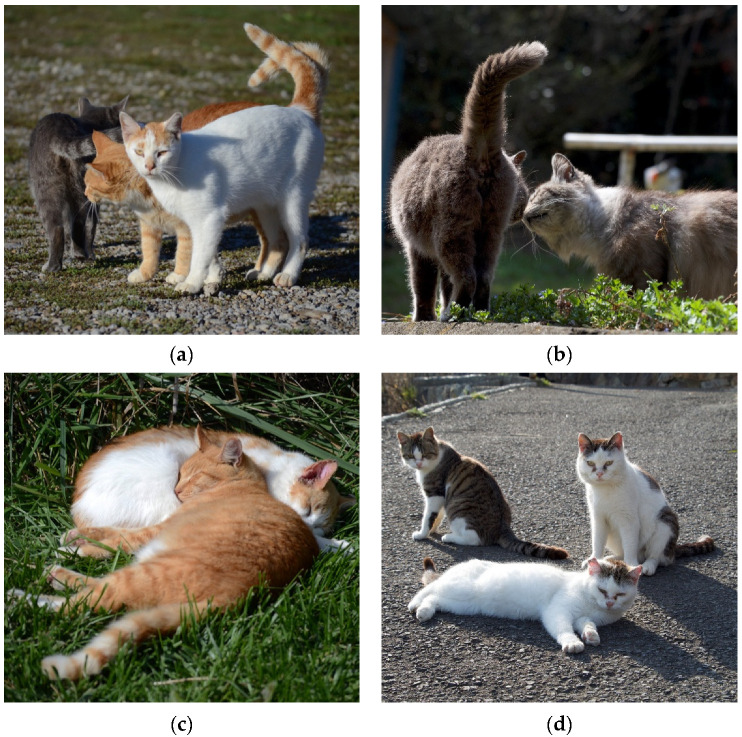
Photos of common FRC social behaviors taken by K. Vitale: (**a**) an orange male cat engages in an allorub with another adult male. Both cats display the tail up signal; (**b**) Two cats engage in a nose sniff, one cat displays tail up; (**c**) two male FRCs sleep together in bodily contact; (**d**) A group of cats sitting in proximity to one another on Tashirojima in Japan.

**Figure 3 animals-12-00126-f003:**
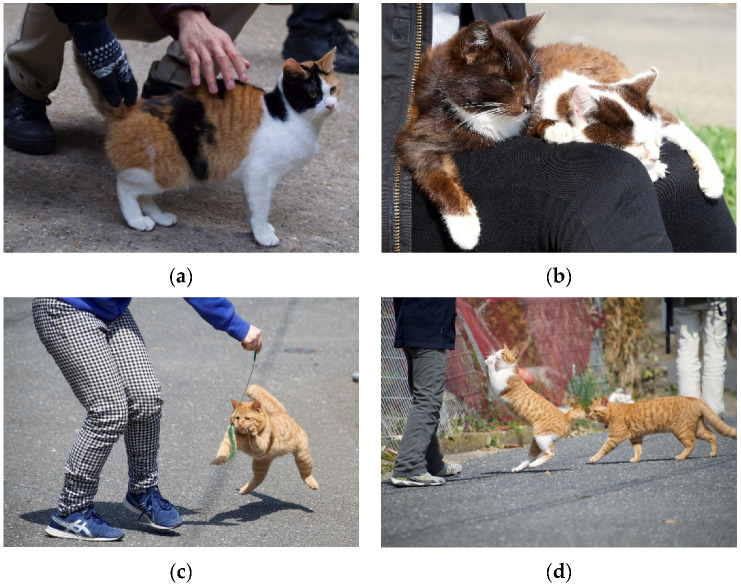
Photos of FRC–human social interactions taken by K. Vitale: (**a**) Cat–human interaction at Fushimi Inari, a shrine in Kyoto, Japan. A FRC accepts petting from multiple people at once; (**b**) FRCs on Tashirojima, an island in Japan, sit on the lap of an unfamiliar human; (**c**) a visitor to Tashirojima plays with a resident cat using a toy; (**d**) another visitor to Tashirojima uses a toy to play with a cat. Other cats start to gather to take turns playing with the toy.

**Table 1 animals-12-00126-t001:** Definitions of key terms related to free-ranging cats (FRCs).

Key Term	Definition	Additional Notes
Domestic Cat	A member of subspecies *Felis silvestris catus*.	Research indicates domestic cats are genetically distinct from their wild ancestors. One gene that was altered during the domestication process was associated with docility (i.e., the ease by which a person can handle/interact with a cat) [7].
Free-Ranging Cat (FRC)	A domestic cat with no constraints on their movement.	A FRC can be socialized or unsocialized (feral). Some FRCs may be lost or abandoned pet cats (unowned strays) and some may be cats that grew up outdoors on a farm (i.e., farm cat) or on the street (i.e., alley cat or street cat) [8].
Socialization	The process by which an individual develops appropriate social behavior.	Socialization, which occurs through experiences with social partners, is necessary for the development of both species specific and interspecific social behavior [9]. Socialization, especially to multiple people, allows cats to learn humans are not to be feared and are safe to approach [10].
Tame (Socialized) Cat	A domestic cat that has been socialized to humans, especially during a sensitive point early in development.	Compared to their wild counterparts, some genetic changes cats experienced during domestication may make it easier to socialize them to humans.
Feral (Unsocialized) Cat	A domestic cat that lacks socialization to humans, especially during a sensitive point early in their development (between the ages of 2–7 weeks [11]).	Cats who lack early experience with humans are often unapproachable and will display fearful, defensive, or aggressive behavior in response to humans [8,12]. The term feral may vary across disciplines. Here it is used to describe the state of an individual [8]. Feral may also be used to describe a population of animals [13]. In this usage, feral animals are formerly domesticated but have undergone significant genetic changes (e.g., through hybridization with wild relatives) such that they are distinct from the domestic population [14,15].
Community Cat	An unowned FRC that is cared for through the cooperation of local residents.	Community cats can be composed of tame and feral individuals. Care may include the implementation of Community Cats Programs (CCPs) or Trap–Neuter–Return (TNR) programs [16].
Cat Colony	A groups of 3 or more adult FRCs that live in close proximity and engage in frequent social behavior.	Slater defined a colony as 3 or more adult cats “living and feeding in close proximity” [8]. Here, the additional criteria of frequent social interaction was added to differentiate bonded social colonies from lose aggregations of cats.

**Table 2 animals-12-00126-t002:** Factors that influence FRC intraspecific social behavior. The factor, behavioral category influenced, and a summary of each influence are provided.

Factors	Behavioral Categories Influenced	Summary of Influence
Sex: Male/Female	Affiliative Caregiving Reproductive	Impact varies by study. Sometimes females are more social, other times males are more social. Both males and females show tolerance for kittens but only females care for kittens. Display of reproductive behavior differs based on sex of conspecifics.
Social Rank: Low/High	Affiliative Aggressive Reproductive	Lower ranked cats display tail up more frequently, higher ranked cats receive tail up more often. Cats of higher social rank display more aggressive behavior. Social rank can impact male reproductive success, although results vary.
Sexual Status: Spayed/Neutered Presence of Estrus Female	Affiliative Aggressive Reproductive	Affiliative behaviors can became more common after neutering however, other work showed some unneutered cats had higher rates of affiliative behavior. Neutered cats can display less aggression than unneutered cats. The presence of an estrus female can impact aggression between males and unreceptive females toward males.
Individuality	Affiliative Reproductive	Some cats tend to initiate affiliative interactions while other cats tend to receive interactions. Individuals display differences in reproductive behavior such as the number of partners courted, duration of courting, and receptivity to mounts. Bold males were found to have the highest reproductive success.
Age: Adult/Kitten	Affiliative Aggressive	Kittens initiated more allorubs than adult cats but the number of initiations decreased as the kitten aged. In one group, the kitten was the most likely individual to initiate social play. Adults of both sexes show infrequent aggression toward kittens.
Group Membership: Intragroup/ Intergroup	Affiliative Aggressive Reproductive	Cats display more affiliative behavior and less aggression toward group members. In contrast, aggression with individuals of other groups is frequent. Some males and females only copulate within their group; however, this behavior can depend on male body size.
Relationship: Kin/Non-kin Familiar/Unfamiliar	Affiliative Caregiving Reproductive	Mother–adult daughter dyads are often seen resting together. Allorubbing is often initiated by mothers to adult daughters. Female cats display more affiliative behavior toward more familiar males. Cats care for their own offspring as well as the offspring of familiar females. FRCs avoid reproduction with close kin.
Food: Present/Absent	Aggressive	Aggressive encounters were infrequent except around food, 97.5% of aggressive encounters occurred near food.
Human Caretaking: Min./Max. Care	Aggressive	Cats with minimal human care displayed higher aggression toward conspecifics than cats that received maximum human care.
Body Weight: Heavy/Light	Reproductive	Heavier males have higher mating success, but results vary. Compared to heavier females with longer cycles, females who were lighter with shorter estrous cycles accepted mounts more frequently.
Heath Status: Presence/Absence FIV+	Reproductive	Males infected with FIV mounted females the most. Socially dominant males tended to be infected by FIV.

**Table 3 animals-12-00126-t003:** Factors influencing FRC interspecific social behavior. The species involved, factors, behavioral category influenced, and a summary of each influence is provided.

	Factors	Behavioral Category Influenced	Notes
FRC–Human	Human Sex: Male/Female Weather: Presence/Absence of Snow and Rain	Affiliative	Considering the interaction between human sex and weather, cats were much friendlier to women on days with bad weather.
FRC–Wildlife	Food: Present/Absent	Non-Predatory Interaction	A significant majority of cat-wildlife interactions occurred within two hours of feeding.

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
