# Peer review of "The Social Lives of Free-Ranging Cats"

_animals, 2022, doi:10.3390/ani12010126_

Round 1

Reviewer 1 Report

This is an interesting review paper on social behavior of Free-ranging cats, with many many social cases in the text. Cat is so common and so uncommon to us, as nearly everyone has ever raised or at least petted with cats, but the behaviors of cat are so unpridictable thus attactive to us. I like these cases in the paper, but as a scientific review paper, it gives too many research details, and loses the scientific questions and future perspective. I suggest the author re-structure the paper, combine similar findings or cases, thus making a much clear and precise review paper. 

Actually, the core theme of the paper is social behavior, then why not categorize the result and discussion according to the behavior type? Right now it is according to location, but as cats are living with humans, and most FRCs are depending to humans, thus the location is not a key factor. And we can find lots of replicated information of social behaviors in current version. A review is not a running account to record every cases, but a summary and forseen of similar findings. So I prefer author to re-organize according to social behaviors, aff, agg, repro, interaction, etc. When dealing with each behavioral category, author then can find what affect the behavior and what how they affect.

I also suggest adding a part of influencing factors of social behavior. Like the debate of if cats have a social life,  of course they have, but obviously cat density is the key influencial factor, when they are living in a low density habitat, probably they are solitary; but when living in a high density, like more than a thousand per km2, of couse they are group-living animals and unavoidably they have to compete, communicate, and exclude, and thus making a complex social cat world. Other factors, like location as author stated, like human cares, like TNRs, etc. would also work on their social life.

There are still several minor comments, like adding a fig to draw whole social life of FRCs, and some others, but as the whole structure needs revised, seems not so important now.

Author Response

Thank you very much for your helpful comments.

I have restructured the paper to summarize the results according to behavior type instead of location. The paper is now reorganized under the headings of behavioral categories (e.g., affiliative, caregiving, agonistic, etc.) I have also condensed the text by combining similar findings and removing replicated information. 

I have also added two additional tables (Tables 2 & 3) which detail the influencing factors on social behavior identified through the review. As stated, in 2.1 Literature Scope, work on cat density was not considered for inclusion in the review, unless the study measured or observed at least one social behavior. For that reason, density did not emerge as an influential factor in the current review. However, I have highlighted additional details on cat density at the end of Study Locations (Section 4.1.1.) and added this as a clear area of future work under Future Directions (Section 4.3.)

Because the text has been condensed, I added additional photos of FRC social behavior to further display the social life of FRCs. However, if the reviewer has future comments on figures I would be happy to hear any additional feedback.

I feel the revisions have greatly improved the quality of the manuscript and I appreciate your time. 

Reviewer 2 Report

This is an excellent, well-needed review conducted in a fair and proper manner. (I only found one minor typing error: l. 803, separate much and variability.)

Author Response

Thank you for your supportive comments. This typing error has been corrected.

Reviewer 3 Report

Well researched and written paper.  The breadth of review will be very helpful to those wanting information about feline social behavior.

Lines 38 & 39 - The word "feral" is technically not used correctly, especially in light of the use of the word "domestic."   Technically a domesticated animal can be tamed or untamed to humans or other species.  "Untamed" would be more appropriate.  "Feral" refers to a group of animals whose ancestors were domesticated and have undergone reverse domestication.  This reviewer recognizes that the "feral" is often applied to FRC's inappropriately, but for such a nice article as this, I would prefer it be corrected to delete "feral" and replace them with "untamed" or "unsocialized to humans."

Author Response

Thank you very much for your supportive and helpful comments.

Your comment on the term feral is a very important one. In order to clarify my terminology I added a new table to the text (Table 1) with definitions of the key terms related to FRCs.

In Table 1, you will see I have left the term feral. I did not feel comfortable removing the term because “feral” was a key phrase used in the literature search (see 2.2. Literature Search & Filtering). It is a valid point that feral can be used to describe a population of animals. However, feral can also be used to denote a trait of an individual and is used synonymously with "unsocialized” or "untamed”. Use of feral to describe lack of socialization is also consistent with the dog literature (Udell & Brubaker, 2016) and human literature (Dombrowski et al., 2011; LaPointe, 2005), in which a feral child is an individual who, from an early age, has grown up in an extended period of isolation with little to no human contact. For these reasons I have left use of the term feral in the manuscript. However, within Table 1 I include a note that use of feral may vary by discipline to ensure readers are clear the term will vary by context.

I hope these revisions are satisfactory and I look forward to hearing additional feedback if there is any.

  • Dombrowski, S. C., Gischlar, K. L., Mrazik, M., & Greer, F. W. (2011). Feral Children. In S. C. Dombrowski, K. L. Gischlar, & M. Mrazik (Eds.), Assessing and Treating Low Incidence/High Severity Psychological Disorders of Childhood (pp. 81–93). Springer.
  • LaPointe, L. L. (2005). Feral children. Journal of Medical Speech Language Pathology, 13(1), vii–ix.
  • Udell, M. A. R., & Brubaker, L. (2016). Are Dogs Social Generalists? Canine Social Cognition, Attachment, and the Dog-Human Bond. Current Directions in Psychological Science, 25(5), 327–333.